# Cornea verticillata and acroparesthesia efficiently discriminate clusters of severity in Fabry disease

**Wladimir Mauhin**[1,2]*, **Olivier Benveniste**[2,3], **Damien Amelin**[2], **Clémence Montagner**[1], **Foudil Lamari**[4,5], **Catherine Caillaud**[6,7], **Claire Douillard**[8], **Bertrand Dussol**[9,10], **Vanessa Leguy-Seguin**[11], **Pauline D'Halluin**[12], **Esther Noel**[13], **Thierry Zenone**[14], **Marie Matignon**[15,16], **François Maillot**[17,18], **Kim-Heang Ly**[19], **Gérard Besson**[20], **Marjolaine Willems**[21], **Fabien Labombarda**[22], **Agathe Masseau**[23], **Christian Lavigne**[24], **Didier Lacombe**[25,26], **Hélène Maillard**[27], **Olivier Lidove**[1,2]

1 Internal Medicine Department, Reference Center for Lysosomal Storage Disorders, Groupe Hospitalier Diaconesses Croix Saint-Simon, Paris, France, 2 UMRS 974, INSERM, Sorbonne Université, Paris, France, 3 Internal Medicine Department, Pitié Salpêtrière University Hospital, Assistance Publique Hôpitaux de Paris, Paris, France, 4 Metabolic Biochemistry Department, Pitié Salpêtrière University Hospital, Assistance Publique Hôpitaux de Paris, Paris, France, 5 Groupe de Recherche Clinique 13 Neurométabolisme, Sorbonne Université, Paris, France, 6 Biochemistry, Metabolomic and Proteomic Department, Necker Enfants Malades University Hospital, Assistance Publique Hôpitaux de Paris, Paris, France, 7 UMRS 1151, INSERM, Institute Necker Enfants Malades, Paris Descartes University, Paris, France, 8 Reference Center for Inborn Metabolic Diseases, Jeanne de Flandres Hospital, Lille, France, 9 Nephrology Department, Assistance Publique Hôpitaux de Marseille, Marseille, France, 10 Centre d'Investigation Clinique 1409, INSERM, Aix Marseille Université, Marseille, France, 11 Internal Medicine and Clinical Immunology Department, Francois Mitterrand Hospital, Dijon, France, 12 Nephrology and Haemodialysis Department, Centre Hospitalier Côte Basque, Bayonne, France, 13 Internal Medicine Department, Strasbourg University Hospital, Strasbourg, France, 14 Internal Medicine Department, Valence Hospital, Valence, France, 15 Nephrology and Renal Transplantation Department, Institut Francilien de Recherche en Néphrologie et Transplantation (IFRNT), Henri-Mondor/Albert-Chenevier University Hospital, Assistance Publique Hôpitaux de Paris, Créteil, France, 16 UMRS 955, Institut Mondor de Recherche Biomédicale, INSERM, University of Paris-Est-Créteil, Créteil, France, 17 Internal Medicine Department, Tours University Hospital, Tours, France, 18 UMRS 1253, University of Tours, Tours, France, 19 Internal Medicine Department, Dupuytren University Hospital, Limoges, France, 20 Neurology Department, Grenoble University Hospital, Grenoble, France, 21 Medical Genetics and Rare Diseases Department, Montpellier University Hospital, Montpellier, France, 22 Cardiology Department, Caen University Hospital, Caen, France, 23 Internal Medicine Department, Hôtel-Dieu University Hospital, Nantes, France, 24 Internal Medicine and Vascular Diseases Department, Angers University Hospital, Angers, France, 25 Medical Genetics Department, Bordeaux University Hospital, Bordeaux, France, 26 INSERM U1211, Bordeaux University, Bordeaux, France, 27 Internal Medicine Department, Huriez Hospital, University of Lille, Lille, France

* wmauhin@hopital-dcss.org

**Data Availability Statement:** All relevant data are within the manuscript and its Supporting Information files.

## Abstract

### Background

Fabry disease (OMIM #301 500), the most prevalent lysosomal storage disease, is caused by enzymatic defects in alpha-galactosidase A (*GLA* gene; Xq22.1). Fabry disease has historically been characterized by progressive renal failure, early stroke and hypertrophic cardiomyopathy, with a diminished life expectancy. A nonclassical phenotype has been described with an almost exclusive cardiac involvement. Specific therapies with enzyme substitution or chaperone molecules are now available depending on the mutation carried.

**Funding:** The author(s) received no specific funding for this work.

**Competing interests:** WM received honoraria, congress fees and travel assistance from Shire-Takeda, Amicus and Sanofi- Genzyme. OB, DA, CM declare no conflict of interest. FLam has received travel support from Amicus Therapeutics., Shire and Sanofi-Genzyme. He received lecture fees from Actelion Pharmaceuticals. BD has received honoraria from Amicus (member of the scientific board) and Novartis (lectures) and travel fees from Genzyme-Sanofi. VLS has received travel fees and accommodations from Shire and Sanofi-Genzyme. CC has received consultant honoraria and congress fees from Biomarin and Sanofi-Genzyme and has participated in editorial activity with Takeda-Shire. CD has received travel assistance from Shire, Sanofi-Genzyme, Sobi, Orphan Europe, Nutricia, Lucane Pharma, Amicus, and Ultragenyx and honoraria from Amicus and has participated on boards with Ultragenyx and Sanofi. OB, AD, PDH declare no conflict of interest EN has received travel fees from Shire and Sanofi-Genzyme and an honorarium from Amicus. TZ has received congress fees and travel assistance from Sanofi-Genzyme. MM declare no conflict of interest FM has received honoraria from Shire and travel assistance from Sanofi-Genzyme. KHL declare no conflict of interestGB has received travel assistance from Shire, Genzyme-Sanofi and Amicus. GB has received travel assistance from Shire, Genzyme-Sanofi and Amicus. MW and FLab declare no conflict of interest AM has received travel fees and accommodations from Shire, Sanofi-Genzyme and Amicus. CL has received honoraria from Sanofi-Genzyme and travel assistance from Sanofi-Genzyme and Shire. DL has received honoraria and travel assistance from Sanofi-Genzyme and has participated on boards with Amicus. HM received honoraria and travel assistance from Sanofi-Genzyme and Amicus and has participated on boards with Amicus and Shire. OL has received travel support and lecture fees from Amicus Therapeutics, Shire, and Sanofi- Genzyme. This does not alter our adherence to PLOS ONE policies on sharing data and materials.

Numerous clinical and fundamental studies have been conducted without stratifying patients by phenotype or severity, despite different prognoses and possible different pathophysiologies. We aimed to identify a simple and clinically relevant way to classify and stratify patients according to their disease severity.

## Methods

Based on data from the *French Fabry Biobank and Registry* (FFABRY; n = 104; 54 males), we applied unsupervised multivariate statistics to determine clusters of patients and identify clinical criteria that would allow an effective classification of adult patients. Thanks to these criteria and empirical clinical considerations we secondly elaborate a new score that allow the severity stratification of patients.

## Results

We observed that the absence of acroparesthesia or cornea verticillata is sufficient to classify males as having the nonclassical phenotype. We did not identify criteria that significantly cluster female patients. The classical phenotype was associated with a higher risk of severe renal (HR = 35.1; p <$10^{-3}$) and cardiac events (HR = 4.8; p = 0.008) and a trend toward a higher risk of severe neurological events (HR = 7.7; p = 0.08) compared to nonclassical males. Our simple, rapid and clinically-relevant FFABRY score gave concordant results with the validated MSSI.

## Conclusion

Acroparesthesia and cornea verticillata are simple clinical criteria that efficiently stratify Fabry patients, defining 3 different groups: females and males with nonclassical and classical phenotypes of significantly different severity. The FFABRY score allows severity stratification of Fabry patients.

## Introduction

Fabry disease (FD; OMIM #301 500) is an X-linked lysosomal storage disease caused by an enzymatic defect of the hydrolase alpha-galactosidase A (AGAL-A), resulting in the accumulation of glycosphingolipids, mainly globotriaosylceramide (Gb3) and its deacetylated form globotriaosylsphingosine (lysoGb3), the latter being commonly used as a surrogate biomarker [1–3]. FD has historically been characterized by acral pain, angiokeratoma, cerebral strokes, progressive renal failure and cardiomyopathy, with a diminished life-expectancy [4]. However, the clinical presentation and incidence of FD are changing as the diagnostic approach is moving from clinicobiochemical algorithms to genetic screenings. Indeed, the first estimations based on clinical ascertainment before 2000 evaluated the incidence of FD between 1:40,000–117,000 live births [5,6], whereas three recent newborn screening studies observed incidences greater than 1:10,000 [7–10]. Since 1990, a nonclassical or late-onset phenotype of FD has been described, with higher residual AGAL-A activity and predominant, if not isolated, cardiac manifestations [11]. The majority of the individuals detected by genetic screenings carry galactosidase A alpha (*GLA*) variants that are usually associated with this nonclassical phenotype of FD [7,12]. The classical and nonclassical phenotypes have been empirically determined

on the basis of the presence or absence of characteristic symptoms (usually neuropathic pain, angiokeratoma, and cornea verticillata (CV)), *GLA* enzyme activity and/or the *GLA* genetic variant, though without any consensus [12,13]. As the prognosis of the different phenotypes is markedly different, there is a need to determine reproducible classification criteria to improve the reliability of therapeutic studies and to personalize the bedside management of FD. Some scoring systems already exist, and they have been elaborated with empirical considerations; these scoring systems include many nonobjective criteria with several items that make them difficult to use in a daily practice. Moreover, existing scoring systems do not differentiate non-classical from classical phenotypes of the disease whereas a growing literature suggests the need for personalized management [14–16]. In this study, we employed unsupervised multi-variate statistics for clinical data to identify simple and objective criteria that would allow an effective classification of adult patients. Additionally, we propose a new and simple scoring system based on this classification to assess the clinical severity and facilitate the management of FD patients.

## Materials and methods

### Patients, clinical data and biological samples

We analyzed data from patients prospectively included in the multicenter cohort FFABRY with an enzymatic and/or genetic diagnosis of FD from December 2014 to May 2017. Written consent were obtained after written and verbal information. The present study was approved by the local ethics committee (Comité de Protection des Personnes VI—Pitié Salpêtrière) and the Comité consultatif sur le traitement de l'information en matière de recherche dans le domaine de la santé, according to the relevant French legislation. Clinical data were prospectively collected through a standardized online form. Cardiac hypertrophy was defined as diastolic interventricular septum thickness $> 13$ mm by cardiac echocardiography or magnetic resonance imaging (MRI). Arrhythmia was defined as the presence of cardiac conduction defect or rhythm trouble. Estimation of the glomerular filtration rate (eGFR) was based on the CKD-EPI equation [17]. Glomerular hyperfiltration was defined as eGFR $> 135$ml/min/$1.73$m$^2$ [18]. Proteinuria was positive if above $0.3$ g/24 h or if the proteinuria/creatininuria ratio was $> 50$ mg/ mmol. Cornea verticillata was assessed via slit-lamp examination. If not mentioned in the medical records, the patients were considered to not have a history of the following items: cerebral stroke, movement disorder, seizure, renal or cardiac transplantation, dialysis, need of a pacemaker (PM), and cardiac failure. All other items were considered missing if not mentioned. The Mainz Severity Score Index (MSSI) was calculated automatically according to the scoring system established by Whybra et al. [19].

Blood samples were collected at the time of inclusion. Plasma was isolated by centrifugation using BD Vacutainer™ serum tubes with an increased silica act clot activator and BD Vacutainer™ heparin tubes before storage at -80˚C. All patients were screened for the presence of anti-agalsidase antibodies, as previously described [20]. LysoGb3 concentrations were measured in available plasma samples (n = 36) by ultra-performance liquid chromatography coupled to tandem mass spectrometry (UPLC-MS/MS), as previously described [20].

### Statistical analyses

Males and females were analyzed separately due to the known phenotype differences [12]. We performed ascending hierarchical clustering on principal components (HCPC) after multiple correspondence analysis (MCA) for the following categorical variables: presence or history of CV, angiokeratoma, history of Fabry acral pain, hypertrophic cardiomyopathy (HCM), arrhythmia, eGFR $</>$ 45 ml/min/1.73 m$^2$, renal transplant, ischemic stroke and hearing loss

and *GLA* variant type (missense *vs.* others). All the categorical variables were used as active and included in the clinical clustering except the *GLA* variant type used as illustrative. Cerebral MRI abnormalities were excluded due to missing data. Age was considered an illustrative variable and was not included in the clustering. MCA and HCPC were performed with R software version 3.4.0 and the package FactoMineR. Patients with missing data were excluded from this analysis. The correlation between variable and dimension was considered significant at $p < 0.02$. We defined the best algorithm to meet the previous clusters using ROC curves. After verification for normal distribution and equality of variances with Shapiro-Wilk and Levene tests, respectively, we employed parametric tests such as the t-test with and without Welch's correction for unequal variances, Pearson correlation and linear regression for Gaussian values, or nonparametric tests such as Kruskal-Wallis (KW) and Mann-Whitney (MW) comparison tests and the Spearman correlation test. We used the log-rank test with Kaplan-Meier to analyze the survival distribution. We applied logistic regression with stepwise selection based on p-values for discrete variables and Fisher's exact t-test for contingency. The p-value for the alpha-risk in all tests, except for MCA and HCPC, was 0.05. GraphPad Prism 5.0 and the EZR plugin version 1.35v [21] packages for R software were used.

## Results

From December 2014 to May 2017, 104 patients (54 males) were prospectively included in the FFABRY cohort. Their general characteristics are described in Table 1.

## Clinical clustering in males

Multiple component analysis (MCA) and hierarchical clustering on principal components (HCPC) were performed with data for 41 male patients who had available complete data (Fig 1). Their mean age was 44.4 years-old. We assume that treatment with ERT or chaperone therapy does not modify the overall phenotype of patients. Six patients were untreated at the inclusion (mean age 34.6 years-old; min–max 17.1–58.1 y.). Mean duration of treatment was 6.5 years in treated patients (min—max: 0.26–15.5 y.). The first 2 dimensions of MCA expressed 53.7% of the total inertia. An ascending HCPC performed on the first 5 dimensions identified 3 different clusters (Fig 2). Group 1 (mean age = 50.5 +/- 11.2 years; n = 21) was characterized by the absence of CV ($p<10^{-7}$), the absence of angiokeratoma ($p<10^{-4}$), the absence of acral pain ($p<0.001$), a missense mutation ($p<0.02$), eGFR > 45 ml/min/1.73 m$^2$ ($p<0.02$), and the absence of renal transplant (p = 0.02). Group 2 (mean age = 32.3 +/- 9.9 years; n = 13) was characterized by the absence of hypertrophic cardiomyopathy (HCM) ($p<10^{-3}$), the presence of CV (p = 0.001), the presence of angiokeratoma (p = 0.003), the presence of acral pain (p = 0.007), and eGFR > 45 ml/min/1.73 m$^2$. Group 3 (mean age = 48.8 +/- 9.5 years; n = 7) was characterized by eGFR < 45 ml/min/1.73 m$^2$ ($p<10^{-6}$), a history of renal transplant ($p<10^{-4}$) and the presence of CV (p = 0.002). On the basis of these characteristics, we considered group 1 to be the nonclassical phenotype and groups 2 and 3 to be the classical phenotype, with younger and older patients, respectively. When considering the absence of acral pain or CV as criteria for the nonclassical phenotype, we met the previously defined clusters with a sensitivity of 89.5%, a specificity of 91.0%, a positive predictive value of 89.5%, a negative predictive value of 91.0%, and an area under the receiver operating characteristic (ROC) curve of 0.9.

## Clinical clustering in females

We applied the same approach for females using the same variables for patients with complete data (n = 36). MCA and HCPC revealed 3 different clusters without any clinical significance (Fig 3). Therefore, we considered that clinical clustering was not appropriate for females.

**Table 1. Characteristics of patients (\*time under enzyme replacement therapy included).**

| Exposure to treatment | Males (n = 54) | | | | | Females (n = 50) | | | | |
|---|---|---|---|---|---|---|---|---|---|---|
| | None | Agalsidase alpha | Agalsidase beta | Agalsidase alpha and beta | Migalastat (+/- agalsidase) | None | Agalsidase alpha | Agalsidase beta | Agalsidase alpha and beta | Migalastat (+/- agalsidase) |
| Exposed (n) | 10 | 12 | 18 | 11 | 3 | 25 | 9 | 8 | 6 | 2 |
| Currently treated (%) | - | 44 (81%) | | | | - | 25 (50%) | | | |
| Median age (Q1Q3) | 27.2 years (20.8–41.0) | 46.1 years (34.3–53.6) | 48.7 years (43.4–60.4) | 42.7 years (31.9–47.8) | 43.3 years (32.7–52.1) | 43.2 years (36.1–53.2) | 52,7 years (48,0–54,6) | 58,5 years (47,6–63,0) | 54,9 years (49,5–61,2) | 52.4 years +/- 8.2 years |
| Median follow-up time (Q1Q3) | 4.1 months (1.9–5.4) | 5.7 years (3.7–7.7) | 7.1 years (2.8–13.8) | 10.7 years (5.2–16.2) | 4.2 years* (3.9–8.7) | 3.0 years (0.6–5.5) | 5.8 years (3.0–12.2) | 14.8 years (11.6–15.9) | 8.6 years (6.2–10.3) | 9.8 years* +/- 8.1 years |
| Median cumulative exposure to specific treatment (Q1Q3) | - | 4.0 years (1.3–6.4) | 4.6 years (0.3–9.3) | 10.6 years (4.0–12.7) | 4.2 years (0.6–8.2)* | - | 2.5 years (2.4–9.9) | 6.7 years (2.2–11.4) | 6.4 years (5.2–10.0) | 2.4 years* +/- 1.6 years |
| Mean age at visit (mean +/- SD) | 43.4 +/- 14.7 years | | | | | 48.9 +/- 14.5 years | | | | |
| Mean +/- SD cumulative exposure to specific treatment | 7.0 +/- 4.8 years | | | | | 6.1 +/- 4.5 years | | | | |
| Median MSSI Neurological [Q1-Q3] (max 20) | 5 [1.3–8] | | | | | 4.5 [1.3–6.0] | | | | |
| MSSI Cardiac (max 20) | 6 [0.0–11.8] | | | | | 2 [0.0–10.8] | | | | |
| MSSI Renal (max 18) | 0 [0–8] | | | | | 0 [0–8] | | | | |
| MSSI General (max 18) | 4 [2–6] | | | | | 2 [1–4] | | | | |
| MSSI Global (max 76) | 20.5 [12.5–28.0] | | | | | 14.5 [7.0–21.8] | | | | |

## FFABRY score

As already mentioned, the morbidity of FD relies on renal, cardiac and central nervous system involvement. Hence, the prognosis of FD depends on the clinical phenotype of patients. Based on the results of the previous clustering, we introduce the first severity scoring system that takes into account the clinical phenotype of FD. The FFABRY score is therefore constructed with 4 variables: the overall clinical phenotype, the kidney disease score, the heart disease score and the central nervous system score as followed:

## Phenotype

- Males with the classical phenotype: past or actual acroparesthesia and cornea verticillata

- Males with the nonclassical phenotype: no past or actual acroparesthesia or no cornea verticillata

- Females

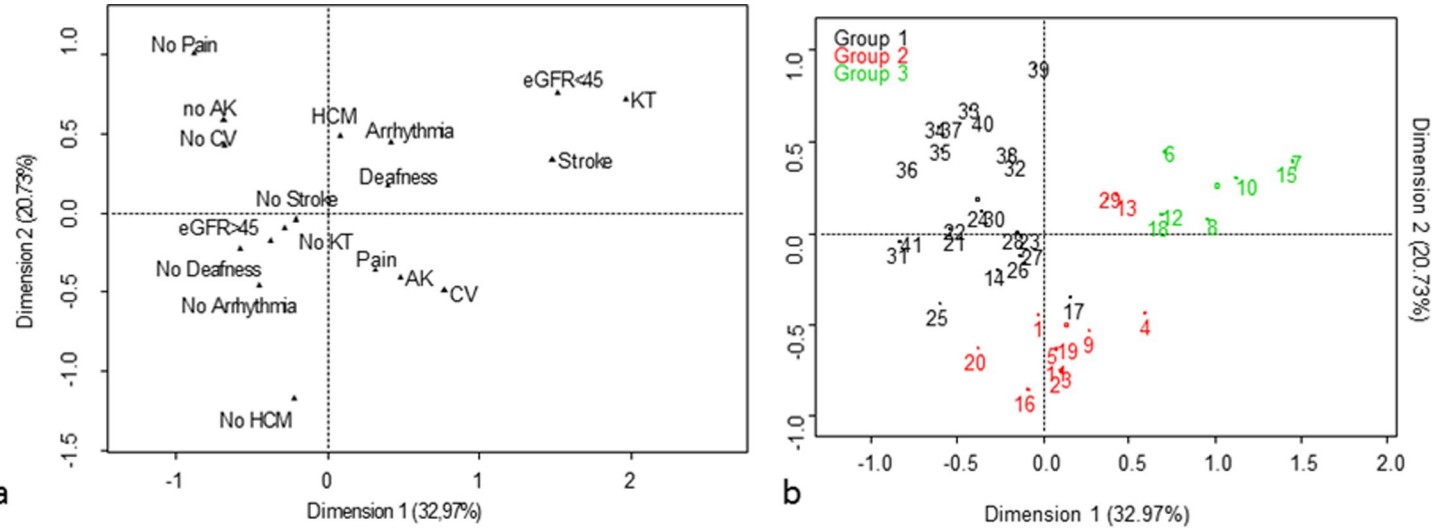

**Fig 1.** a: Variable factor map obtained by multiple component analysis using data for 41 males with complete data (AK: angiokeratoma; CV: cornea verticillata; eGFR: estimated glomerular filtration rate in ml/min/1.73 m$^2$; HCM: hypertrophic cardiomyopathy; KT kidney transplantation). b: Ascending hierarchical classification of individuals using the first 5 dimensions of multiple component analysis performed using data for 41 males with complete data. Three clusters were identified. Group 1, characterized by the absence of cornea verticillata (p<10$^{-7}$), the absence of angiokeratoma (p<10$^{-4}$), the absence of acral pain (p<0.001) and the absence of renal disease, was referred to as the nonclassical cluster. Groups 2 and 3, characterized by the presence of cornea verticillata (p<0.002), were referred to as classical groups with younger (mean age 32.3 +/- 9.9 years) and older (mean age 48.8 +/- 9.5 years) patients, respectively.

**Kidney disease: K score from K0 to K5.**  FD renal involvement is progressive and characterized by glomerular hyperfiltration and proteinuria followed by a decrease in glomerular function [22,23]. We propose the scale as follows:

K0: No proteinuria and 90 > eGFR > 135 ml/min/1.73 m$^2$

K1: Hyperfiltration such as eGFR $\geq$ 135ml/min/1.73m$^2$ without proteinuria

K2: 60 < eGFR $\leq$ 90 ml/min/1.73 m$^2$ OR proteinuria > 0.3g/24h or 50 mg/ mmol

K3: 30 < eGFR $\leq$ 60 ml/min/1.73 m$^2$ +/- proteinuria.

K4: 15 < eGFR $\leq$ 30 ml/min/1.73 m$^2$ +/- proteinuria.

K5: eGFR $\leq$ 15 ml/min/1.73 m$^2$ or dialysis or renal transplant.

**Heart disease: H score from H0 to H4.**  FD is a cause of HCM, progressively leading to diastolic dysfunction, ischemia or obstructive cardiac failure [24]. FD is also characterized by arrhythmia, which has become the leading cause of death [25,26]. An interventricular septum thickness (IST) > 30 mm has been associated with a high risk for sudden death [24]. Additionally, we propose the following staging:

H0: No HCM. No cardiac symptomatology.

H1: HCM such as 13 < IST $\leq$ 30 mm and/or QRS interval on ECG $\geq$ 200 msec and/or ventricular hypertrophy on ECG without cardiac symptoms and no need of antiarrhythmic or beta-blockers.

H2: H1 + need of antiarrhythmic or beta-blockers*.

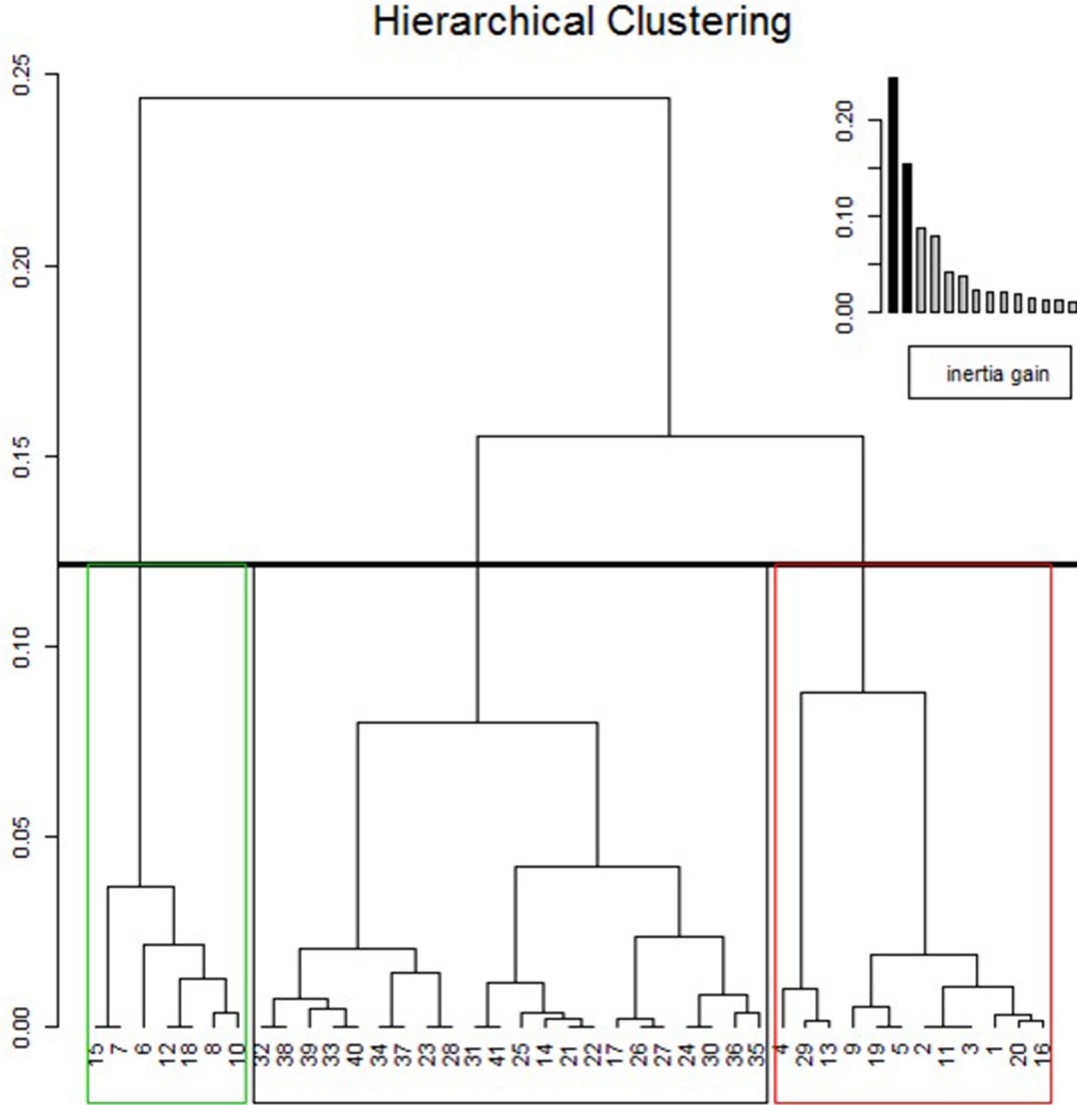

**Fig 2. Hierarchical clustering on the first 5 dimensions of the multiple component analysis performed on data of the 41 males with complete data.**

H3: 35% < Left ventricular ejection function (LVEF) ≤ 50% and/or a need of pacemaker (PM) implantation and/or angina.

H4: Need of heart transplant or LVEF ≤ 35% or IST > 30 mm.

* according to guidelines on management of HCM [27]

**Central nervous system involvement: N score from N0 to N2.** In a logistic regression model, we observed that chronic headaches were associated with a higher risk of stroke in males (OR 16.2, p = 0.01), independent of renal and cardiac diseases. Moreover, a trend toward a higher risk of cerebral stroke was observed in males with cochlear disorder defined by the presence of tinnitus or hearing loss (OR 7.8, p = 0.054), independent of age. Additionally, we propose the following staging:

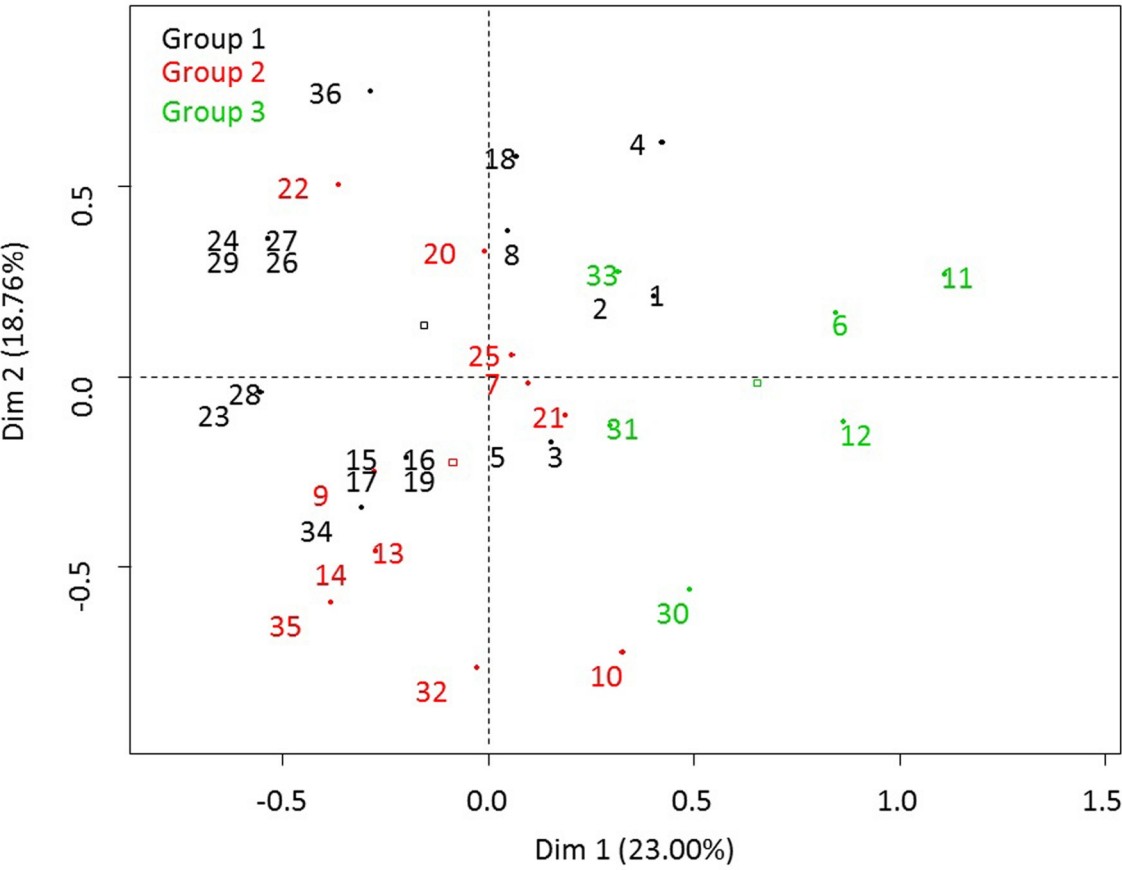

**Fig 3. Ascending hierarchical classification of individuals (using the first 5 dimensions of multiple component analysis performed using data for 36 females with complete data) did not reveal any significant groups.**

N0: No chronic headache and no cochlear disorder and no history of stroke.

N1: Chronic headache and/or tinnitus and/or hearing loss and no history of stroke.

N2: History of cerebral stroke.

**Total FFABRY score from T0 to T11.** The total (T) score was defined as the sum of the K-, H- and N-scores.

## Description of the FFABRY cohort clustered by sex, cornea verticillata and acroparesthesia

Using our clustering criteria, we distinguished 29 males with the classical phenotype, 25 males with the nonclassical phenotype and 50 females. Their clinical and biochemical characteristics are described extensively in Table 2. Briefly, among males, those with the nonclassical phenotype were diagnosed at an older age (45.4 vs. 28.6 years; $p < 10^{-3}$) and had a milder phenotype including a less steep eGFR slope (-1.7 vs. -2.4 ml/min/1.72m$^2$/ years; $p < 0.05$), a lower risk of renal transplantation (log-rank, HR = 0.07; $p < 10^{-3}$) and a lower risk of cerebral stroke (1/25 vs. 4/29; non-significant). Surprisingly, ENT involvement was frequent in both male groups, being observed in 70.0% of classical and 52.0% of nonclassical cases (p = ns). Of note, one of the most commonly reported symptoms was anxiety, which was observed in 70.3% of the male

**Table 2. Characteristics of patients stratified by clinical phenotype defined with the FFABRY scoring system.**

| Phenotype | Classical Males | Nonclassical Males | Females |
|---|---|---|---|
| n | 29 | 25 | 50 |
| Median age [IQR] (years) | 39.8 [29.5–46.3] | 51.8 [43.0–60.8] | 51 [38.0–58.5] |
| Median age at diagnosis (years) | 28.6 | 45.4 | 42.8 |
| Treatment received (N =) | 25 | 19 | 25 |
| Agalsidase alpha (n =) | 8 | 4 | 9 |
| Agalsidase beta (n =) | 8 | 10 | 8 |
| Both agalsidases successively (n =) | 8 | 3 | 6 |
| Migalastat (n =) | 1 | 2 | 2 |
| Mean cumulative duration of treatment (+/- SD; years) | 10.3 +/- 8.5 | 13.1 +/- 5.1 | 6.1 +/- 4.0 |
| **Renal disease** | | | |
| R-score > 0 | 15 | 19 | 24 |
| Median R-score | 1 | 0.5 | 0 |
| Median renal MSSI (IQR) | 4 (0–18) | 0 (0–8) | |
| eGFR slope in ml/min/1.73 m²/y (r²; p) | -2.4 (0.66; p < 0.0001) | -1.7 (0.61; p < 0.0001) | -0.7 (0.19; p < 0.003) |
| ESRD (n) | 8[1] | 0 | 1 |
| Median survival without renal transplantation | 48.6 years | NA | NA |
| ACE blocker (n) | 14 | 14 | 16 |
| ACE blockers in patients with R-score >0 | 67.9% | | 50% |
| **Cardio-vascular disease** | | | |
| HCM[2] (n =) | 15 | 21[3] | 19 |
| Median survival without HCM | 46.3 years | 57.6 years [4] | 62.0 years |
| Pacemaker (n =) | 1 | 6 [5] | 1 |
| Median survival without severe cardiac event (H-score ≥ 3) | 50.8 years | 60.8 years[6] | 71.2 years |
| **Neurological disease** | | | |
| Dyshidrosis (n) | 18/29 | 6/25 (p = 0.005) | 34/50 |
| Heat intolerance (n) | 17/23 | 8/23 (p = 0.008) | 9/40 |
| Ischemic strokes (n; %) | 4; 13.8% | 1; 4.0% | 9; 18.0% |
| Median age [IQR]; years | 27.6 [15.0–33.4] | 34.0 | 45.1 [45.1–61.9] |
| **ENT involvement** | | | |
| Tinnitus | 12/22 (54.5%) | 7/22 (31.8%) | 15/44 (34.1%) |
| Hearing loss | 17/24 (70.8%) | 13/25 (52.0%) | 20/48 (41.7%) |
| **Ophthalmological involvement** | | | |
| Cataract (median age at diagnosis; years) | 5/12 (44.5) | 4/20 (64.3) | 6/42 (60.1) |
| Cornea verticillata | 19/19 | 0/23 | 28/42 |
| **Other** | | | |
| Angiokeratoma | 23 (79.3%) | 8 (33.3%)[7] | 18/47 (38.3%) |
| Abdominal pain | 14/27 (51.8%) | 5/17 (29.4%) | 8/42 (24.4%) |
| **Mental health** | | | |
| Anxiety | 12/38 (70.3%) | | 16/34 (47.1%) |
| Depression symptoms | 11/38 (28.9%) | | 15/44 (44.1%) |
| Suicide attempt | 4/38 | | 0 |
| Mutations | c.137A>G* | c.334C>T* | c.123del* |
| | c.169C>T | c.337T>C* | c.125T>G* |
| | c.233C>G | c.486G>C* | c.214del (n = 2)* |
| | c.334C>T (n = 3)* | c.522T>A* | c.233C>G (n = 2) |
| | c.424T>C | c.644A>G (n = 9)* | c.334C>T (n = 5)* |
| | c.486G>C | c.692A>G* | c.424T>C* |

*(Continued)*

**Table 2.** (Continued)

| | | |
|---|---|---|
| c.539del* | c.713G>A(n = 2)* | c.427G>A (n = 2)* |
| c.548G>C* | c.758T>C | c.486G>C* |
| c.680G>A | c.802-3_802-2del* | c.504A>C* |
| c.729G>C* | c.847C>T (n = 2)* | c.548G>C* |
| c.798T>A (n = 2)* | c.902G>A* | c.655A>C* |
| c.802-3_802-2del* | c.1010T>C* | c.680G>A* |
| c.806T>C* | c.1016T>G* | c.695T>C* |
| c.847C>T* | c.1087C>T (n = 2)* | c.718_719del* |
| c.875C>T | | c.729G>C* |
| c.884T>G | | c.798T>A |
| c.901C>T (n = 2)* | | c.802-3_802-2del (n = 4)* |
| c.902G>A* | | c.840A>T |
| c.1010T>C* | | c.884T>G (n = 3)* |
| c.1069_1079del* | | c.901C>T (n = 6)* |
| c.1246C>T* | | c.902G>A* |
| no data (n = 4) | | c.1277_1278del (n = 2)* |
| | | c.1021del |
| | | c.1024C>T |
| | | c.1087C>T |
| | | c.1168G>A |
| | | no data (n = 6) |
| **LysoGb3 (median in ng/ml; IQR; n)** | | |
| **In treated patients** | 18.9 (11.6–32.3; n = 17) | 6.25 (2.6–21.9; n = 17)[8] | 4.5 (2.7–6.2; n = 15) |
| **In untreated patients** | 101.8 (n = 2) | 8.5 (3.0–16.7; n = 5) | 2.6 (1.7–3.8; n = 22) |

eGFR: estimated glomerular filtration rate; ESRD: end-stage renal disease; ACE-blocker: angiotensin conversion enzyme blocker; HCM: hypertrophic cardiomyopathy; PM: pacemaker; NA: not available. (1) HR vs. nonclassical 14.4; log-rank p = 0.0003; (2) in Cox regression, HCM is influenced by age at diagnosis (HR 0.81; p<10−7) and cumulative exposure to treatment (HR 0.71; p < 0.0001); (3) in Cox regression, influenced by the phenotype: HR nonclassical vs. classic: 0.19; p = 0.006); (4) log-rank classical vs. nonclassical; p < 0.003; (5) log-rank classical vs. nonclassical; p = ns; (6) log-rank classical vs. nonclassical; p = 0.01; (7) Fisher-exact t-test classical vs. nonclassical; p<0.001; (8) Mann-Whitney, p = 0.01

*variants included in the MCA and HCPC analyses (exhaustive data). One 46.6-year-old female, a p.A143T carrier, was included in the MCA and HCPC analyses. She had been treated for 2 years with migalastat for acral and abdominal pain and angiokeratoma but had no renal, cardiac or cerebrovascular involvement (MSSI = 13; plasma lysoGb3 1.1nM). Another female p.A143T carrier, aged 43.0 years, had no dermatological, ophthalmological, neurological, renal or cardiological symptoms (MSSI = 1; plasma lysoGb3 1.3nM). She received no specific treatment and was not included in the MCA and HCPC analyses due to missing data.

patients, with suicide attempts reported for 10.5% of the male cohort. Females exhibited a contrasting globally milder phenotype (Fig 4 and Table 2). However, 18.0% of them had a history of ischemic stroke, with a median age at diagnosis of 45.1 years, and the event was significantly associated with the existence of cardiac rhythm problems (OR: 15.3; p = 0.046). Such an association with rhythm problems was not observed in men.

## Performances of clustering with MSSI and FFABRY scores in the entire FFABRY cohort

Clustering appeared clinically relevant using both MSSI and FFABRY scores, with significant differences between groups stratified by classes of ages (Table 3). The classical phenotype was associated with a higher risk of severe renal events (K-score ≥ 3; Cox analysis: HR (classical/ nonclassical) = 35.1; p <10−3; HR (classical/ females) = 100; p < 10−5; Fig 4) and a higher risk of severe cardiac events (H-score ≥ 3; Cox analysis: HR (classical vs. nonclassical) = 4.8;

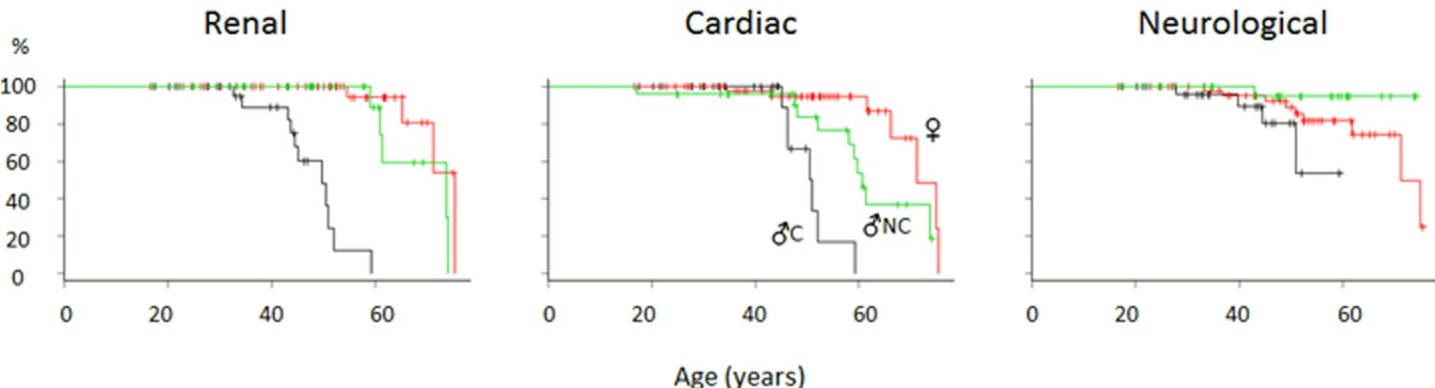

**Fig 4. Renal, cardiac and neurological severe event-free survival curves (black: Classical group males, green: Nonclassical group males, red: Females).**

p = 0.008; HR (classical/ females) = 16.7; $p < 10^{-4}$; Fig 4), and there was a trend toward a higher risk of severe neurological events (N-score = 2) in males with the classical compared to the nonclassical phenotype (Cox analysis; HR = 7.7; p = 0.08, Fig 4). There was no significant difference among females (HR = 2.5; p = 0.2). T-score evolution illustrated that the overall severity increased with time in the classical group compared to the nonclassical group (Fig 5).

### LysoGb3 plasma levels, anti-agalsidase antibodies and FFABRY

LysoGb3 plasma levels were higher in males with the classical phenotype compared to those with the nonclassical phenotype or females in both treatment-naïve (respective medians 101.8, 5.8 and 3.2; Kruskal Wallis p < 0.04) and in treated (respective medians 18.9, 6.7 and 4.5; $p < 10^{-4}$) patients. As expected, there was no correlation between lysoGb3 plasma levels and FFABRY scores after stratification by phenotype among treated patients. Two female and 18 male patients were positive for anti-agalsidase antibodies. If considering only males who had been exposed to agalsidase, the presence of antibodies was significantly associated with the classical phenotype (56% vs. 20%, p < 0.02). Their characteristics have already been reported [20].

### Genotype-phenotype correlation

Among the 30 different genetic variants observed in males, nonsense variants were associated with the classical phenotype (9/13 patients with nonsense variants). The four patients classified in the nonclassical group had no cornea verticillata: a c.802-3_802-2del carrier with a K1H3N0 score at 17.1 years, two c.847C>T carriers with K0H1N0 and K0H1N1 scores at 33.4 and 47.3 years, and a c.522T>A carrier with a K2H0N0 score at 47.7 years. Five variants were observed in both the classical and nonclassical groups: c.334C>T, c.847C>T, c.802-3_802-2del, c. 902G>A and c.1010T>C. Their characteristics are described in Table 4.

### Discussion

The nonclassical phenotype of FD has become the most prevalent [7,12]. However, most clinical and fundamental research studies addressing with FD have not stratified patients based on phenotype. Here, we propose a simple algorithm to distinguish patients. We demonstrate that CV assessed by slit-lamp and acral pain, but not angiokeratoma, are statistically sufficient to

**Table 3.** FFABRY and MSSI scores (comparison with Mann-Whitney test p*: Classical versus nonclassical males; p**: Males versus females).

| | Males | | | Females | |
|---|---|---|---|---|---|
| | **Classical** | **Nonclassical** | **p*** | | **p**** |
| n | **29** | **25** | | **50** | |
| **Age total** | **39.84 [29.49, 46.29]** | **51.76 [42.96, 60.76]** | **0.001** | **51.00 [37.99, 58.51]** | **0.001** |
| < 35 y n (median [IQR]) | 14 (28.69 [21.69, 32.51]) | 6 (29.20 [24.90, 34.35]) | 0.509 | 10 (27.16 [25.09, 32.51]) | 0.766 |
| 35–50 y n (median [IQR]) | 11 (44.50 [43.52, 46.30]) | 6 (47.46 [44.28, 47.66]) | 0.228 | 13 (43.14 [38.39, 46.56]) | 0.233 |
| > 50 y n (median [IQR]) | 4 (51.58 [50.93, 53.93]) | 13 (60.76 [58.10, 67.58]) | 0.024 | 27 (58.22 [52.95, 65.26]) | 0.074 |
| **FFABRY Heart score** | **1.00 [0.00, 2.00]** | **2.00 [1.00, 3.00]** | **0.085** | **1.00 [0.00, 2.00]** | **0.025** |
| < 35 y (median [IQR]) | 0.00 [0.00, 1.00] | 1.00 [0.25, 1.75] | 0.130 | 0.00 [0.00, 0.00] | 0.213 |
| 35–50 y (median [IQR]) | 1.00 [1.00, 2.50] | 1.50 [0.25, 2.75] | 0.716 | 0.00 [0.00, 2.00] | 0.113 |
| > 50 y (median [IQR]) | 3.50 [3.00, 4.00] | 3.00 [2.00, 3.00] | 0.040 | 2.00 [1.00, 2.00] | 0.001 |
| **FFABRY Kidney score** | **1.00 [0.00, 5.00]** | **0.50 [0.00, 2.00]** | **0.214** | **0.00 [0.00, 2.00]** | **0.228** |
| < 35 y (median [IQR]) | 0.00 [0.00, 0.75] | 0.00 [0.00, 0.75] | 1.000 | 0.00 [0.00, 0.00] | 0.902 |
| 35–50 y (median [IQR]) | 2.00 [1.00, 5.00] | 0.00 [0.00, 0.00] | 0.024 | 1.00 [0.00, 2.00] | 0.026 |
| > 50 y (median [IQR]) | 4.50 [4.00, 5.00] | 2.00 [1.50, 3.00] | 0.004 | 2.00 [0.00, 2.00] | 0.004 |
| **FFABRY Neurological score** | **1.00 [0.00, 1.00]** | **1.00 [0.00, 1.00]** | **0.340** | **1.00 [0.00, 1.00]** | **0.669** |
| < 35 y (median [IQR]) | 0.50 [0.00, 1.00] | 0.00 [0.00, 0.00] | 0.165 | 1.00 [0.00, 1.00] | 0.231 |
| 35–50 y (median [IQR]) | 1.00 [0.25, 1.00] | 1.00 [0.25, 1.00] | 0.859 | 0.00 [0.00, 1.00] | 0.743 |
| > 50 y (median [IQR]) | 1.00 [1.00, 1.25] | 1.00 [0.00, 1.00] | 0.076 | 1.00 [0.00, 1.00] | 0.311 |
| **FFABRY Total score** | **3.00 [1.00, 7.00]** | **4.00 [2.00, 6.00]** | **0.882** | **2.50 [1.00, 4.75]** | **0.312** |
| < 35 y (median [IQR]) | 1.00 [1.00, 1.75] | 1.50 [1.00, 2.00] | 0.724 | 1.00 [0.00, 1.00] | 0.487 |
| 35–50 y (median [IQR]) | 5.50 [3.25, 7.00] | 2.50 [2.00, 3.75] | 0.043 | 2.00 [1.00, 4.25] | 0.030 |
| > 50 y (median [IQR]) | 9.00 [8.00, 10.25] | 6.00 [4.00, 6.00] | 0.003 | 3.00 [2.00, 5.00] | 0.002 |
| **MSSI cardiac** | **2.00 [0.00, 9.00]** | **10.00 [3.00, 14.00]** | **0.013** | **2.00 [0.00, 10.75]** | **0.025** |
| < 35 y (median [IQR]) | 0.50 [0.00, 2.75] | 4.50 [0.00, 9.00] | 0.505 | 0.00 [0.00, 1.50] | 0.348 |
| 35–50 y (median [IQR]) | 6.00 [2.00, 10.50] | 6.50 [1.50, 9.25] | 0.724 | 0.00 [0.00, 2.00] | 0.047 |
| > 50 y (median [IQR]) | 1.50 [0.00, 6.75] | 14.00 [13.00, 16.00] | 0.190 | 9.00 [1.50, 14.00] | 0.067 |
| **MSSI general** | **5.00 [3.00, 8.00]** | **2.00 [1.00, 4.00]** | **0.001** | **2.00 [1.00, 4.00]** | **<0.001** |
| < 35 y (median [IQR]) | 4.50 [2.00, 6.00] | 2.00 [1.25, 2.00] | 0.054 | 1.50 [1.00, 4.75] | 0.167 |
| 35–50 y (median [IQR]) | 5.00 [4.00, 8.50] | 5.50 [3.25, 7.00] | 0.577 | 1.00 [1.00, 3.00] | 0.005 |
| > 50 y (median [IQR]) | 6.50 [5.00, 9.25] | 1.00 [1.00, 2.00] | 0.006 | 2.00 [1.50, 4.00] | 0.006 |
| **MSSI renal** | **4.00 [0.00, 18.00]** | **0.00 [0.00, 8.00]** | **0.052** | **0.00 [0.00, 8.00]** | **0.111** |
| < 35 y (median [IQR]) | 0.00 [0.00, 0.00] | 0.00 [0.00, 0.00] | 0.342 | 0.00 [0.00, 0.00] | 0.558 |
| 35–50 y (median [IQR]) | 8.00 [6.00, 18.00] | 0.00 [0.00, 0.00] | 0.015 | 4.00 [0.00, 4.00] | 0.010 |
| > 50 y (median [IQR]) | 13.00 [8.00, 18.00] | 8.00 [0.00, 8.00] | 0.017 | 0.00 [0.00, 8.00] | 0.030 |
| **MSSI neurological** | **7.00 [3.00, 10.00]** | **2.00 [0.00, 5.00]** | **0.001** | **4.50 [1.25, 6.00]** | **0.002** |
| < 35 y (median [IQR]) | 6.00 [2.25, 8.00] | 4.00 [2.25, 5.75] | 0.534 | 5.50 [5.00, 7.50] | 0.640 |
| 35–50 y (median [IQR]) | 6.00 [3.50, 12.00] | 4.00 [0.75, 7.25] | 0.362 | 5.00 [1.00, 6.00] | 0.348 |
| > 50 y (median [IQR]) | 9.00 [7.75, 10.75] | 1.00 [0.00, 2.00] | 0.004 | 3.00 [0.00, 5.50] | 0.004 |
| **MSSI total** | **24.00 [14.00, 33.00]** | **20.00 [12.00, 24.00]** | **0.080** | **14.50 [7.00, 21.75]** | **0.017** |
| < 35 y (median [IQR]) | 14.00 [8.75, 18.75] | 10.50 [4.75, 18.50] | 0.535 | 11.00 [5.50, 18.50] | 0.704 |
| 35–50 y (median [IQR]) | 29.00 [24.50, 32.50] | 16.50 [12.00, 24.75] | 0.039 | 13.00 [7.00, 15.00] | 0.002 |
| > 50 y (median [IQR]) | 35.50 [33.75, 37.25] | 22.00 [18.00, 24.00] | 0.005 | 19.00 [9.00, 26.00] | 0.014 |

distinguish males with the classical phenotype from those with the nonclassical phenotype, with significant differences in terms of severity. As expected, males with the classical phenotype have more severe renal disease than do males with the nonclassical phenotype, but the

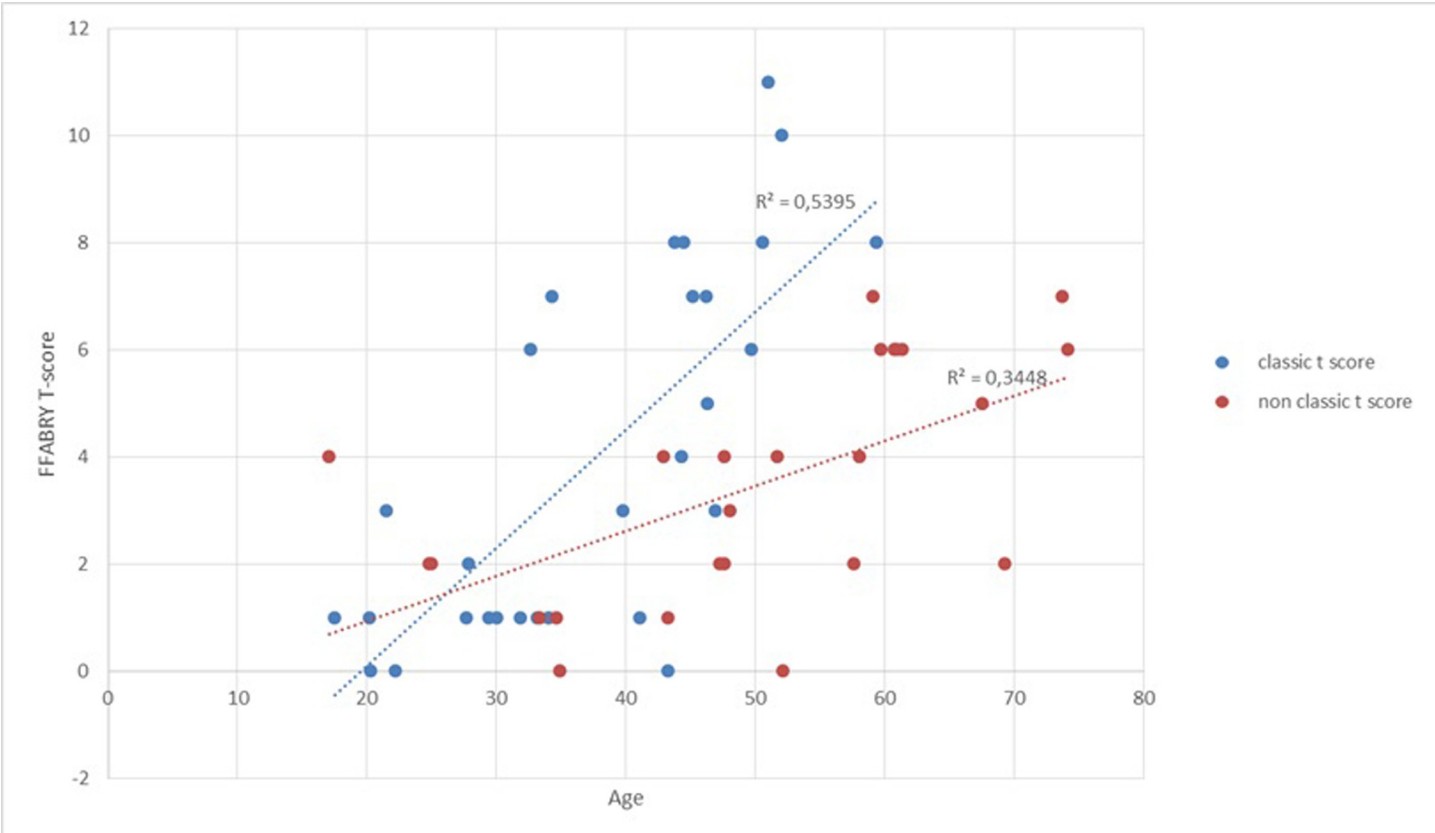

**Fig 5. T-score evolution in the classical group compared to the nonclassical group (with associated linear regression curve).**

former also experience cardiomyopathy earlier, and some of them also have stroke early, independent of arrhythmia. Females cannot be separated into classical and nonclassical phenotypes, likely due to the X-inactivation status in the different organs.

Hemangiomas, which are frequent, can easily be misdiagnosed as angiokeratoma, which may explain the reduced specificity of angiokeratoma for the classical phenotype. In contrast, CV is a pivotal criterion in our classification. Although CV can be related to exposure to amiodarone, its association with the severity of FD has already been described, as observed in 94% of classical Fabry cases [28]. A recent paper reported that CV is often underdiagnosed in FD

**Table 4. Characteristics of patients with genetic variants observed in both classical and nonclassical groups (eGFR: Estimated glomerular filtration by CKD-EPI equation in ml/min/1.73m$^2$; RT: Renal transplant).**

| Variant | Classical patient | | | | | Nonclassical patient | | | |
|---|---|---|---|---|---|---|---|---|---|
| | age | Treatment duration (y) | FFABRY score | eGFR | | age | Treatment duration (y) | FFABRY score | eGFR |
| c.334C>T | 34.4 | 6.2 | K5 H1 N1 T7 | RT | | 48.1 | 13.4 | K0 H3 N0 T3 | 105 |
| | 45.2 | 3.4 | K3 H3 N1 T7 | 52 | | | | | |
| | 59.4 | 5.6 | K4 H3 N1 T8 | 23 | | | | | |
| c.802-3_802-2del | 50.6 | 1.3 | K4 H3 N1 T8 | 25 | | 17.1 | 0 | K1 H3 N0 T4 | 136 |
| c.847C>T | 49.8 | 11.1 | K5 H1 N0 T6 | RT | | 33.4 | 13.3 | K0 H1 N0 T1 | 116 |
| | | | | | | 47.3 | 13 | K0 H1 N1 T2 | 102 |
| c.902G>A | 17.5 | 0.4 | K1 H0 N0 T1 | 136 | | 24.9 | 0 | K0 H2 N0 T2 | 121 |
| c.1010T>C | 46.3 | 4.3 | K2 H3 N0 T5 | 81 | | 60.9 | 4.8 | K3 H2 N1 T6 | 57 |

using the slit-lamp approach [29]. The present work concerned 4 males whose phenotype was not mentioned and 10 females. However, whether a more sensitive approach using in vivo corneal confocal microscopy (IVCM) may reveal minimal deposits in nonclassical patients and females is not the question. Hence, we suggest that an unquestionable CV, with obvious deposits assessed by slit-lamp, is associated with a classical FD phenotype. Similarly, our second pivotal criterion is the existence past or actual of acral pain and not the proof of small fiber neuropathy assessed by paraclinical exams.

Our study was based on data for 104 adult patients; a small number of patients is inherent with the rarity of FD. However, FFABRY is a multicenter database including patients from different pedigrees, different medical specialties (nephrology, cardiology, internal medicine, and genetics) and different locations in France, which allows for clinical and genetic heterogeneity (42 different variants) that benefits analyses. We used a linear regression model to assess the slope of eGFR evolution in the cohort, which is limited by the heterogeneity of patients, especially females. Analysis of X-inactivation may allow for a better classification of the female phenotype and help in stratifying the risk of severe events; unfortunately, this analysis remains unavailable in routine practice [30]. As experts managing lysosomal diseases in a dedicated tertiary center, we assume that treatment with ERT or chaperone therapy has not modified the overall natural history of the disease or the prognosis of patients. Nonetheless, to the best of our knowledge, regression but no disappearance of CV has ever been reported [31].

In the era of evidence-based medicine, severity scores have become mandatory for evaluating therapeutics. With FFABRY, we propose a clinically and statistically objective severity scoring system for FD. The FFABRY score was developed based on the natural history of FD and the severe clinically relevant events we observed in our center of expertise for lysosomal diseases. Our N-score highlights the importance of cochlear disorders and headaches, which are associated with the risk of cerebral stroke in males, whereas white matter lesions (WMLs) were not related to any specific symptom.

FFABRY allowed us to establish a portrait of current Fabry patients, distinguishing males with classical and nonclassical phenotypes and females with their proper clinical specificity. Interestingly, we observed no systematic genotype-phenotype correlation. Some patients sharing the same genetic variant were classified into two different groups. The FFABRY scores as well as individual clinical criteria demonstrated that our classification performed better than genotype for describing disease severity. Other scoring systems have already been developed for FD. The MSSI, which has been the most commonly used scoring system, was developed on the basis of data from 24 males and 15 females registered in the Fabry Outcome Survey (FOS) database (Shire-Takeda)[19]. The MSSI includes 26 variables, empirically weighted, among which hemorrhoids, facial appearance and subjective fitness assessment are notable; however, the MSSI does not include renal transplantation. The Disease Severity Scoring System (DS-3), which was elaborated by experts from the Fabry Registry (Sanofi-Genzyme), also includes nonobjective items such as the "patient reported domain" or sweating capacity. WMLs are included, though they are asymptomatic and not associated with poorer outcomes. Finally, the different items and their weightings have been empirically and not statistically established in these two scoring systems. Moreover, the number of items makes them challenging to use and time consuming. The Fabry International Prognostic Index (FIPI) appears to be much more robust, as it has been developed on the basis of multivariate analyses of data from 1483 patients from the FOS registry [32]. Although FIPI appears to be an effective prediction tool, it does not allow assessment of the actual clinical severity. Indeed, five of the six variables included in the cardiac item refer to extracardiac symptoms (eGFR, proteinuria, deafness, vertigo, and angiokeratoma). The online Fabry Stabilization index (FASTEX) was recently validated [33], and the authors established an attractive tool for personal follow-up of individuals.

Nevertheless, clinical phenotypes are not distinguished in this system, which could be misleading for interindividual comparisons, making the scoring system useless in group studies. After stratification according to phenotype, FFABRY scores allowed a rapid and clinically relevant evaluation of disease severity. Analyses of the K score highlight the suboptimal management of FD females in our cohort. Our study also emphasizes windows of opportunity for the diagnosis and introduction of specific treatments, as follows: before 30 years old in males with the classical phenotype, 45 years old in males with the nonclassical classical phenotype and 50 years old in females. Regarding the obviously different prognoses observed with the FFABRY score, we believe that the classical and nonclassical phenotypes should be considered as two different subtypes of FD in males, with different management strategies. As for Niemann-Pick disease A and B or Gaucher types 1, 2 and 3, it would be useful to rename the phenotypes for male patients, with classical as type 1 and nonclassical as type 2, to fully take into account such obvious clinical differences and possible pathophysiological differences.

Identifying acral pain and cornea verticillata is a rapid and simple approach that statistically discriminates Fabry phenotypes.

## Supporting information

**S1 Data. Clinical and biological data of included patients.**
(XLSX)

## Acknowledgments

We gratefully thank the Société Nationale Française de Médecine Interne and Vaincre les maladies lysosomales patient association for their support as well as Isabelle Citerne and Epiconcept's team for their help.

## Author Contributions

**Conceptualization:** Wladimir Mauhin, Olivier Lidove.

**Data curation:** Wladimir Mauhin, Olivier Benveniste, Damien Amelin, Clémence Montagner, Foudil Lamari, Catherine Caillaud, Claire Douillard, Bertrand Dussol, Vanessa Leguy-Seguin, Pauline D'Halluin, Esther Noel, Thierry Zenone, Marie Matignon, François Maillot, Kim-Heang Ly, Gérard Besson, Marjolaine Willems, Fabien Labombarda, Agathe Masseau, Christian Lavigne, Didier Lacombe, Hélène Maillard, Olivier Lidove.

**Formal analysis:** Wladimir Mauhin.

**Investigation:** Wladimir Mauhin, Clémence Montagner, Foudil Lamari, Catherine Caillaud, Claire Douillard, Bertrand Dussol, Vanessa Leguy-Seguin, Pauline D'Halluin, Esther Noel, Thierry Zenone, Marie Matignon, François Maillot, Kim-Heang Ly, Gérard Besson, Marjolaine Willems, Fabien Labombarda, Agathe Masseau, Christian Lavigne, Didier Lacombe, Hélène Maillard, Olivier Lidove.

**Methodology:** Wladimir Mauhin, Olivier Benveniste, Damien Amelin, Olivier Lidove.

**Project administration:** Wladimir Mauhin, Olivier Benveniste.

**Supervision:** Wladimir Mauhin.

**Validation:** Wladimir Mauhin, Olivier Benveniste.

**Writing – original draft:** Wladimir Mauhin, Olivier Benveniste, Damien Amelin, Foudil Lamari, Catherine Caillaud, Olivier Lidove.

**Writing – review & editing:** Wladimir Mauhin, Olivier Benveniste, Damien Amelin, Clémence Montagner, Foudil Lamari, Catherine Caillaud, Claire Douillard, Bertrand Dussol, Vanessa Leguy-Seguin, Pauline D'Halluin, Esther Noel, Thierry Zenone, Marie Matignon, François Maillot, Kim-Heang Ly, Gérard Besson, Marjolaine Willems, Fabien Labombarda, Agathe Masseau, Christian Lavigne, Didier Lacombe, Hélène Maillard, Olivier Lidove.

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
