## [Decision Letter · Decision Letter 0]

24 Mar 2020

PONE-D-19-35853

Cornea verticillata and acroparesthesia efficiently discriminate clusters of severity in Fabry disease

PLOS ONE

Dear Dr MAUHIN,

Thank you for submitting your manuscript to PLOS ONE. After careful consideration, we feel that it has merit but does not fully meet PLOS ONE’s publication criteria as it currently stands. Therefore, we invite you to submit a revised version of the manuscript that addresses the points raised during the review process.

We would appreciate receiving your revised manuscript by May 08 2020 11:59PM. To enhance the reproducibility of your results, we recommend that if applicable you deposit your laboratory protocols in protocols.io, where a protocol can be assigned its own identifier (DOI) such that it can be cited independently in the future. For instructions see: http://journals.plos.org/plosone/s/submission-guidelines#loc-laboratory-protocols

We look forward to receiving your revised manuscript.

Kind regards,

Maria Vittoria Cubellis

Academic Editor

PLOS ONE

Additional Editor Comments (if provided):

Dear authors,

your paper

Cornea verticillata and acroparesthesia efficiently discriminate clusters of severity in Fabry disease

is not acceptable for publication in the present state.

I am ready to reconsider your paper if you take into account reviewers' suggestions

Journal Requirements:

2. Please provide additional details regarding participant consent. In the ethics statement in the Methods and online submission information, please ensure that you have specified (1) whether consent was informed and (2) what type you obtained (for instance, written or verbal, and if verbal, how it was documented and witnessed).

3. Thank you for including your ethics statement: "- Comité de Protection des Personnes VI (Pitié Salpêtrière) authorization date: 04-03-2014

- Comité consultatif sur le traitement de l’information en matière de recherche dans le domaine de la santé (CCTIRS n°14.324bis of the 06-18-2014)

Written consent"

a) Please amend your current ethics statement to confirm that your named institutional review board or ethics committee specifically approved this study.

4. Your ethics statement must appear in the Methods section of your manuscript. If your ethics statement is written in any section besides the Methods, please move it to the Methods section and delete it from any other section. Please also ensure that your ethics statement is included in your manuscript, as the ethics section of your online submission will not be published alongside your manuscript.

"WM received honoraria, congress fees and travel assistance from Shire-Takeda, Amicus and Sanofi-Genzyme.

FLam has received travel support from Amicus Therapeutics., Shire and Sanofi-Genzyme. He received lecture fees from Actelion Pharmaceuticals.

OL has received travel support and lecture fees from Amicus Therapeutics, Shire, and Sanofi-Genzyme.

DL has received honoraria and travel assistance from Sanofi-Genzyme and has participated on boards with Amicus.

HM received honoraria and travel assistance from Sanofi-Genzyme and Amicus and has participated on boards with Amicus and Shire.

BD has received honoraria from Amicus (member of the scientific board) and Novartis (lectures) and travel fees from Genzyme-Sanofi.

VLS has received travel fees and accommodations from Shire and Sanofi-Genzyme.

EN has received travel fees from Shire and Sanofi-Genzyme and an honorarium from Amicus.

AM has received travel fees and accommodations from Shire, Sanofi-Genzyme and Amicus.

CC has received consultant honoraria and congress fees from Biomarin and Sanofi-Genzyme and has participated in editorial activity with Takeda-Shire.

TZ has received congress fees and travel assistance from Sanofi-Genzyme.

FM has received honoraria from Shire and travel assistance from Sanofi-Genzyme.

CL has received honoraria from Sanofi-Genzyme and travel assistance from Sanofi-Genzyme and Shire.

CD has received travel assistance from Shire, Sanofi-Genzyme, Sobi, Orphan Europe, Nutricia, Lucane Pharma, Amicus, and Ultragenyx and honoraria from Amicus and has participated on boards with Ultragenyx and Sanofi.

GB has received travel assistance from Shire, Genzyme-Sanofi and Amicus.

OB, AD, PDH, KHL, FLab, MM, CM, and MW declare no conflict of interest."

Reviewers' comments:

Reviewer's Responses to Questions

**Comments to the Author**

1. Is the manuscript technically sound, and do the data support the conclusions?

Reviewer #1: Partly

Reviewer #2: Yes

2. Has the statistical analysis been performed appropriately and rigorously? 

Reviewer #1: I Don't Know

Reviewer #2: Yes

3. Have the authors made all data underlying the findings in their manuscript fully available?

Reviewer #1: Yes

Reviewer #2: Yes

4. Is the manuscript presented in an intelligible fashion and written in standard English?

Reviewer #1: Yes

Reviewer #2: Yes

5. Review Comments to the Author

Reviewer #1: Many thanks for submitting the manuscript PONE-D-19-35853 for review. The authors Mauhin and colleagues present an informative study that aims to clinically classify Fabry patients. The authors develop a new scoring system to classify FD patients and conclude that acral pain (experienced in the past or presence) and cornea verticillata suffice to discriminate classical from nonclassical FD patients.

The strength of the study lies in the very well documented patient cohort of 104 male and female FD patients from various centers in France. The evaluations and statistical tests used are, in the opinion of this reviewer, correctly chosen.

Nevertheless, I think that the study is quite ambitious and in the current version confusing at times. I will specify my criticism in the following points so that the authors can make changes if necessary, because I think that clarity needs to be improved in order to reach a broader readership.

Major points:

1. Abstract: In general, and this should perhaps be addressed in the abstract, I am unsure about the message and merit of the study. In the abstract the "methods" describe very briefly the statistical evaluation. Would it not be more appropriate to mention the approach of introducing a new score to validate the patient clusters obtained. I think that would summarize the article better. Is a completely new score even suitable to validate the clusters? Or would you maybe call it a “comparison”. As I said I am quite unsure what to do with this approach even though I think it is interesting. Since you use also the approved MSSI score for "validation" or correlation of the data (Table 3), maybe this can be named here as well. Since this should be described in more detail, the incredibly detailed "Results" section could then be shortened by a few of the given numbers and only mention general findings. Detailed results should then be named in the actual results section.

2. Introduction: A further question of understanding arises from the last sentence of the introduction, page 7, lines 122-ff. How do the authors transmit from the clinical clustering to the FFABRY score? What is the relationship between the score, as strangely enough introduced in the results section starting on page 13, and the clustering data shown above (Figures 1 and 2)? If I understand it correctly, clustering is being conducted qualitatively, binary, yes/no, but the score suddenly comes up with 6 K-classes for kidney disease, 5 H classes for heart disease and 3 N classes for neurological involvement. Of course I understand the introduction of a scoring system that is as precise as possible. The scores of the parameters are added to form a total score, but how does this fit to the clustering approach into basically 2(!) patient groups (or 3, if I consider the females) introduced before? I feel a logical disruption here, 2 different subjects being mixed up. The score results from the existing patient data and the clustering also, but should the score result from the clustering data either? That connection is missing, especially since all 54 male individuals obtain an FFABRY score, but were not used for the clustering. The title of Table 2 states: “Characteristics of patients stratified by clinical phenotype defined with the FFABRY scoring system.” If the FFABRY score is exclusively used to classify the patients, there should not be an overlap in FFABRY scores in the classical and nonclassical group. In this case, I think it is better to indicate the min-max than the IQR, because IQR confusingly suggests an intersection of the groups.

3. Results: About the clustering criteria. If I understand this correctly, each criterion can be answered in binary form. Acral pain/no acral pain, CV/no CV and so on. Since it is mentioned in the text as a distinguishing criterion for the nonclassical group 1 (line 182) I searched for "no missense mutation" or similar in figure 1a, but I did not find it. Maybe it helps to sum up all criteria in a table.

4. Results: Table 3 contains a lot of information, but is hardly mentioned or explained in the text. You just state (page 18, line 300-f.): “Clustering appeared clinically relevant using both MSSI and FFABRY scores, with significant differences between groups stratified by classes of ages (table 3).“ However, the statistical assessment was carried out between the 3 groups classical, nonclassical and female. Here is the question, what do we learn from this? Statistical significance rarely seems to be the case here. If age is a significant factor, then surely statistics should have been displayed between age groups within each patient group? That confuses me a bit?

5. Discussion: I would be interested how many patients in this cohort were initially classified as classical before the score was applied? Did the score obtained match the original assessment? In this regard it is important to note whether the physical examinations for the renal, cardiac and neurological parameters have been performed by the same physician in all 104 patients? And was the MSSI score taken as a basis? As you write, it is subject to a certain amount of subjectivity.

6. Discussion: I am unsure about this point (it relates to point 5 above), but are the numerical thresholds used for the FFABRY score objective? As can be seen from the biomarker, patients with higher lyso-Gb3 levels are not necessarily more severely affected. In other words, how objectively can one say that a patient with 60 < eGFR ≤ 90 ml/min/1.73 m² is less severely affected than a patient with 30 < eGFR ≤ 60 ml/min/1.73 m²? This should also be discussed I think. In the end, the overall condition of the patient always tells us how severely he/she is affected or am I wrong?

7. Discussion: What is the overall benefit to use the Score developed in this study? Maybe you could give examples to clarify this. I give an example: A study that discovers a missense mutation that is amenable to chaperone therapy has a direct impact on a patient with such mutation. The clinical phenotype of the patient plays a minor role. The main indicator is the knowledge about the genetics of the patient. I mean, certainly in nonclassical cases a decision on therapy will be approached more cautiously than in classical cases, where the start of therapy must not be delayed. But does this study contribute anything to that? Don't get me wrong, I think the new score is very informative and objectifies the issue somehow, maybe it should be mentioned in the title of the study? But as you write, there is hardly a novelty in the finding that CV discriminates the Fabry patient groups, I quote from the article (page 23, line 350-ff.): "Although CV can be related to exposure to amiodarone, its association with the severity of FD has already been described, as observed in 94% of classical Fabry cases (22)". In other words, isn´t there any evidence that the classification can be helpful in therapy decisions as to start an ERT or PCT for example? This topic should be discussed.

8. Figures: Basically all figures, but especially Figure 4 require strong editing. The statements made cannot be reconstructed at all on the basis of the figure, as the caption is not legible. The legend for Figure 4 does not provide much information either. I would recommend the use of panel "A", "B", and "C" for renal, cardiac, and neurological. Do the numbers in Figure 3 have a meaning? They are not readable. Do they denote patient IDs as in Figure 1b? Figure 5 reads the French term "Linéaire". This should be translated into English.

Minor points:

Page 4, l. 75: GLA gene should be written in italics (holds true throughout the text)

Page 6, l. 106-ff.: The sentence on page 6, starting at line 106 reports 3 recent newborn studies. First, the references for the low incidence should be placed after "live births [3,4],..." and the references [5-7] should be placed as is. However, the authors omit a fourth study by Wittmann and colleagues (JIMD Rep. 2012;6:117-25. doi: 10.1007/8904_2012_130.), which reaches a similar result in a newborn study in Hungary and should also be cited here.

Page 12, line 177-ff.: It is not indicated whether the 41 subjects for this analysis also included untreated patients. Apart from the split between the sexes, I consider this to be an essential point to consider. Even though this following statement is made on page 24, line 368-ff.: “As experts managing lysosomal diseases in a dedicated tertiary center, we assume that treatment with ERT or chaperone therapy has not modified the overall natural history of the disease or the prognosis of patients. Nonetheless, to the best of our knowledge, regression but no disappearance of CV has ever been reported (25)." This must be introduced directly at this point, as it otherwise contributes to the confusing structure of the article.

Page 16, line 267: Should not the 41 males and 36 females with complete data be used for this analysis? Would the result change?

Page 23, line 366-ff. and page 24, line 371-ff. Redundant information is given.

Reviewer #2: The manuscript Cornea verticillata and acroparesthesia efficiently discriminate clusters of severity in

Fabry disease by Wladimir MAUHIN and coworkers, concerns with Fabry disease and specifically with the search for “a simple and clinically relavent way to classify patients according to their disease severity”.

Overall, the manuscript is well organized and described.

I think that it deserves to be published. I would suggest few minor revisions.

1 Line 100-102: The authors state: “Fabry disease (FD; OMIM #301 500) is an X-linked lysosomal storage disease caused by an enzymatic defect of the hydrolase alpha-galactosidase A (AGAL-A), resulting in the accumulation of glycosphingolipids, mainly globotriaosylceramide (Gb3) (1)”.

Please shortly cite and discuss LysoGb3. Despite the fact that Gb3 accumulates, LysoGb3 is the biomarker usually measured, in fact the authors show and discuss results concerning LysoGb3 (for example see the paragraph “LysoGb3 plasma levels, anti-agalsidase antibodies and FFABRY” and table2).

The following references could be useful:

Smid, B.E.; van der Tol, L.; Biegstraaten, M.; Linthorst, G.E.; Hollak, C.E.; Poorthuis, B.J. Plasma globotriaosylsphingosine in relation to phenotypes of Fabry disease. J. Med. Genet. 2015, 52, 262–268.

Young-Gqamana, B.; Brignol, N.; Chang, H.H.; Khanna, R.; Soska, R.; Fuller, M.; Sitaraman, S.A.;Germain, D.P.; Giugliani, R.; Hughes, D.A.; et al. Migalastat hcl reduces globotriaosylsphingosine (lyso-Gb3) in Fabry transgenic mice and in the plasma of Fabry patients. PLoS ONE 2013, 8, e57631

2 Lines 119-120. The authors state: “As the prognosis of the different phenotypes is markedly different, there is a need to determine reproducible classification criteria to improve the reliability of therapeutic studies and to personalize the bedside management of FD. Some scoring systems already exist, and they have been elaborated with empirical considerations; these scoring systems include many nonobjective criteria with several items that make them difficult to use in a daily practice”.

It would be useful to introduce some references. For example:

Fabry disease revisited: Management and treatment recommendations for adult patients. Ortiz A, Germain DP, Desnick RJ, Politei J, Mauer M, Burlina A, Eng C, Hopkin RJ, Laney D, Linhart A, Waldek S, Wallace E, Weidemann F, Wilcox WR. Mol Genet Metab. 2018 Apr;123(4):416-427. doi: 0.1016/j.ymgme.2018.02.014. Epub 2018 Feb 28. Review.

The Large Phenotypic Spectrum of Fabry Disease Requires Graduated Diagnosis and Personalized Therapy: A Meta-Analysis Can Help to Differentiate Missense Mutations. Citro V, Cammisa M, Liguori L, Cimmaruta C, Lukas J, Cubellis MV, Andreotti G. Int J Mol Sci. 2016 Dec 1;17(12). pii: E2010. Review.

Long Term Treatment with Enzyme Replacement Therapy in Patients with Fabry Disease. Oder D, Nordbeck P, Wanner C. Nephron. 2016;134(1):30-6. doi: 10.1159/000448968. Epub 2016 Aug 27. Review.

Fabry disease: Review and experience during newborn screening. Hsu TR, Niu DM. Trends Cardiovasc Med. 2018 May;28(4):274-281. doi: 10.1016/j.tcm.2017.10.001. Epub 2017 Oct 20. Review.

3 Table 3. Please specify the meaning of “p”

6. PLOS authors have the option to publish the peer review history of their article (what does this mean?). If published, this will include your full peer review and any attached files.

Reviewer #1: No

Reviewer #2: No

---

## [Author Response · Author response to Decision Letter 0]

4 May 2020

Response to Editor

#1 We revised documents to fulfill with PLOS ONE’s style requirement

#2 & #3 : We added/ modified informations in the manuscript and the online submission information. In the manuscript now appears “Written consent were obtained after written and verbal information. The present study was approved by the local ethics commitee (Comité de Protection des Personnes VI - Pitié Salpêtrière) and the Comité consultatif sur le traitement de l’information en matière de recherche dans le domaine de la santé, according to the relevant French legislation. »

#4 Our ethics statement appears in the Methods section of the manuscript.

#5 We have inquired about updated conflicts of interests. No modification has been done. Also we confirm and mention that they do “not alter our adherence to PLOS ONE policies on sharing data and materials”

#6. We had caption of the supporting information file at the end of the manuscript

Answers to reviewers:

Reviewer #1

#1 We first thank the reviewer for the constructive comments on our manuscript. We have taken in account the feeling of the reviewer about the lack of clarity. We modified the abstract to this end, notably by mentioning the new score we introduce in the manuscript. Nevertheless, we could not explicit the whole statistical protocol in the abstract and invite readers to refer to the method section due to the lack of space. We had to shorten the “Results” part as proposed by the reviewer to allow the modifications. We propose the following modifications:

In the “Methods” section: Thanks to these criteria and empirical clinical considerations we secondly elaborate a new score that allow the severity stratification of patients.

In the “Results”section: The classical phenotype was associated with a higher risk of severe renal (HR = 35.1; p <10-3) and cardiac events (HR = 4.8 ; p = 0.008) and a trend toward a higher risk of severe neurological events (HR = 7.7; p = 0.08) compared to nonclassical males. Our simple, rapid and clinically-relevant FFABRY score gave concordant results with the validated MSSI.

#2 We understood this remark. As mentioned in the introduction, some facts have to be noticed: 

1. “the prognosis of the different phenotypes is markedly different” 

2. “there is a need to determine reproducible classification criteria” for phenotype

3. “scoring systems already exist” but they do not differentiate nonclassical from classical phenotypes of the disease. Finally they “include many nonobjective criteria with several items that make them difficult to use in a daily practice”. 

Hence, existing scores cannot be used to stratify patients. 

We aimed first to easily classify the clinical phenotype of patients and performed hierarchical clustering to this end. Secondly, we aimed to elaborate a simple score that take into account this previous classification in order to stratify patients by groups of comparable prognosis. The phenotype classification is the starting point and the fundamental characteristic of the FFABRY score. 

We modified the text to specifically explicit this point.

If I understand it correctly, clustering is being conducted qualitatively, binary, yes/no, but the score suddenly comes up with 6 K-classes for kidney disease, 5 H classes for heart disease and 3 N classes for neurological involvement. Of course I understand the introduction of a scoring system that is as precise as possible. The scores of the parameters are added to form a total score, but how does this fit to the clustering approach into basically 2(!) patient groups (or 3, if I consider the females) introduced before? I feel a logical disruption here, 2 different subjects being mixed up. The score results from the existing patient data and the clustering also, but should the score result from the clustering data either? That connection is missing, especially since all 54 male individuals obtain an FFABRY score, but were not used for the clustering. 

We propose the FFABRY score as an extension of the statistical work that should not be considered as a mix up. As mentioned in the introduction we propose a “scoring system based on this [phenotype] classification to assess the clinical severity” of patients. The prognosis of Fabry disease is related to kidney failure, cardiac hypertrophy and cerebral stroke. Therefore, we proposed unquestionable empirical criteria of severity in terms of renal, cardiac and central nervous system involvement based on the existent literature. Criteria are well defined in the manuscript. The interest and the novelty of our FFABRY score lie on the phenotype classification defined by the previous statistical clustering. It allows an unbiased stratification of severity, organ by organ, according to the clinical phenotypes. The FFABRY score has not been introduced to validate our statistical work. It has been introduced to serve as a new simple tool to stratify Fabry disease patients.

We have added 2 sentences to introduce the specific paragraph in order to clarify this point: “As already mentioned, the morbidity of FD relies on renal, cardiac and central nervous system involvement. Hence, the prognosis of FD depends on the clinical phenotype of patients. Based on the results of the previous clustering, we introduce the first severity scoring system that takes into account the clinical phenotype of FD. The FFABRY score is therefore constructed with 4 variables: the clinical phenotype, the kidney disease score, the heart disease score and the central nervous system score as followed: …”

The title of Table 2 states: “Characteristics of patients stratified by clinical phenotype defined with the FFABRY scoring system.” If the FFABRY score is exclusively used to classify the patients, there should not be an overlap in FFABRY scores in the classical and nonclassical group. In this case, I think it is better to indicate the min-max than the IQR, because IQR confusingly suggests an intersection of the groups.

There may be a misunderstanding: the clinical phenotype is a fundamental variable of the scoring system. We have tried to correct this misunderstanding with the introduction added to the score paragraph: “The FFABRY score is therefore constructed with 4 variables: the overall clinical phenotype, the kidney disease score, the heart disease score and the central nervous system score as followed:..”. 

We definitely think that IQR is the most effective illustration of dispersion in such non-normal Gaussian distributions.

#3. It is more complicated. Multiple correspondence analysis allows to analyze categorical variables expressed in binary form. The list is already mentioned in the Method section “multiple correspondence analysis (MCA) for the following categorical variables: presence or history of CV, angiokeratoma, history of Fabry acral pain, hypertrophic cardiomyopathy (HCM), arrhythmia, eGFR </> 45 ml/min/1.73 m², renal transplant, ischemic stroke, hearing loss and GLA variant type (missense vs. others).” Each variable can be active, participating into the dispersion map, or illustrative, not included in the calculation: clusters do not take into account illustrative variables. However, illustrative variables can be used to describe clusters.

Among the categorical variables, the GLA variant type (missense vs. others) was considered as an illustrative variable because we aimed to elaborate a clinical clustering. Age was also considered as an illustrative variable because of the bias it could introduce. Therefore, we can describe the prevalence of missense mutations and the age distribution in clusters but these variables were not weighted in the calculation and therefore did not appear on the dispersion map figure 1. We add the following sentence in the Method section to clarify: “All the categorical variables were used as active and included in the clinical clustering except the GLA variant type used as illustrative”

#4. First of all, yes, the age is a fundamental factor in Fabry disease. Fabry disease is a degenerative disease in which older patients are necessarily more severe. As already described in the methods, the age was used as an illustrative variable in the statistics because including it as an active variable would have led to mix not yet severe young classical patients and non severe old nonclassical patients. This is illustrated in the figure 1 in which groups 2 and 3 share common characteristics such as cornea verticillata being considered as classical patients, but mean ages of the groups are different, the youngest being the less severe. 

Including the age as an active variable for clustering could have been possible in a very large cohort that is impossible for Fabry disease that remains a rare disease.

The table 3 illustrates first the comparability of the validated but very time consuming MSSI score and our simple FFABRY score. Also, it shows the importance of considering both ages and clinical phenotypes to stratify patients. 

#5. The FFABRY cohort was initiated in 2014. It gathers data from 17 tertiary centers in France. The physical examinations were performed locally by different practitioners, guided with a standardized form that does not mention the clinical phenotype. There was therefore no classification before the clustering. Although empirically validated, the MSSI score takes into account a lot of (very) subjective (“characteristic facial appearance”, fatigue, depression, diaphoresis) and some inconsistent variables such as the presence of hemorrhoids… With FFABRY, we aimed to elaborate a score based on statistics and strong, objective and reproducible evaluation.

It is important to note that the phenotype is not defined by the score itself but by the absence or presence of cornea verticillata or acral pain. The FFABRY score is supplementary tool, based on this classification and elaborate to stratify patients taking into account these clinical phenotypes.

#6. We may not understand your question. The clinical phenotype is essential, especially for young patients. The presence of cornea verticillata and/or acral pain will lead us to be more aggressive in the treatment and surveillance than for other patients. Once more, the clinical phenotype classification is different from the score.

This is what we wanted to illustrate in the discussion with the sentence “Regarding the obviously different prognoses observed with the FFABRY score, we believe that the classical and nonclassical phenotypes should be considered as two different subtypes of FD in males, with different management strategies.”

However, your remark seems to address to a larger problem than the one of the FFABRY score but to the validity of thresholds in general. Also, we agree that a patient with an eGFR of 58ml/min.1.73m² is not so different from another with 63ml/min/1.73m²… The fact is that we try to elaborate a reproducible stratifying score, and we need thresholds to do so. The different thresholds used for the FFABRY score have been chosen on the basis of already validated scales such as the international KDIGO classification for renal disease or on existing literature.

The FFABRY score is not elaborated to manage patients at bedside. It has not been elaborated to substitute the overall clinical evaluation. It has been elaborated to stratify patients mainly in clinical research to allow the comparison of comparable patients.

#7. We may have answered this point in the previous one at some point.

We agree that there is no individual benefit for the patient to be evaluated with the FFABRY score, no more than with the MSSI or the FIPI. The clinical phenotype is essential, especially for young patients. The presence of cornea verticillata and/or acral pain will lead us to be more aggressive in the treatment and surveillance than for other patients. 

Once more, the clinical phenotype classification is different from the score. Also, the FFABRY score has been elaborated to stratify patients mainly for clinical research to allow the comparison of comparable patients. As we observe such different prognoses between phenotypes, it is essential to stratify patients upon clinical phenotypes to evaluate the different therapeutical strategies properly. Secondly, we already know that the preexistence of organ involvement, such as proteinuria or cardiac fibrosis, before treatment is determinant for the overall prognosis. That is what we try to illustrate with our sentence:

“clinical phenotypes are not distinguished in [the existing scoring systems], which could be misleading for interindividual comparisons, making the scoring system useless in group studies. After stratification according to phenotype, FFABRY scores allowed a rapid and clinically relevant evaluation of disease severity.”

#8. We edited figures 3, 4 and 5 to improve readability. 

Numbers in Fig3 referred to IDs. We modified the numbers as they appear in the database to improve readability. It does not change the meaning of the figure that clustering is not relevant for women patients.

We have moved the fig4 to the paragraph “Performances of clustering with MSSI and FFABRY scores in the entire FFABRY cohort » where it appears more appropriate and comprehensible. 

Minor points:

Page 4, l. 75: GLA gene should be written in italics (holds true throughout the text)

>> We modified all the occurrences. 

Page 6, l. 106-ff.: The sentence on page 6, starting at line 106 reports 3 recent newborn studies. First, the references for the low incidence should be placed after "live births [3,4],..." and the references [5-7] should be placed as is. However, the authors omit a fourth study by Wittmann and colleagues (JIMD Rep. 2012;6:117-25. doi: 10.1007/8904_2012_130.), which reaches a similar result in a newborn study in Hungary and should also be cited here.

>> Thank you for this remark. We did the modifications and added the reference.

Page 12, line 177-ff.: It is not indicated whether the 41 subjects for this analysis also included untreated patients. Apart from the split between the sexes, I consider this to be an essential point to consider. Even though this following statement is made on page 24, line 368-ff.: “As experts managing lysosomal diseases in a dedicated tertiary center, we assume that treatment with ERT or chaperone therapy has not modified the overall natural history of the disease or the prognosis of patients. Nonetheless, to the best of our knowledge, regression but no disappearance of CV has ever been reported (25)." This must be introduced directly at this point, as it otherwise contributes to the confusing structure of the article.

>> Thank you for this remark. As you observed we already mentioned in the discussion that we assumed that the effect of treatment is not sufficient to modify the overall prognosis of patients. In order to improve clarity and quality, we have added to the text the following sentences : “Their mean age was 44.4 years-old. We assume that treatment with ERT or chaperone therapy does not modify the overall phenotype of patients. Six patients were untreated at the inclusion (mean age 34.6 years-old). Mean duration of treatment was 6.5 years in treated patients.“ 

Page 16, line 267: Should not the 41 males and 36 females with complete data be used for this analysis? Would the result change?

>>Indeed it would have improved the description of the cohort. Nevertheless, in the table 2, we excluded missing data for each corresponding item as mentioned by denominator. We have decided to include all the patients in the description of the cohort because in such rare diseases we think that patient with incomplete data can bring important information without introducing bias. 

Page 23, line 366-ff. and page 24, line 371-ff. Redundant information is given.

Thank you for this remark, indeed we think that X-inactivation analysis would be very very interesting. We corrected this point.

Reviewer #2: 

#1 We thank the reviewer 2 for his review and his remarks. We modified the manuscript to introduce the lysoGb3 as a surrogate biomarker in Fabry disease. Although we used it, we did not go into further details because the role of biomarker is still debated. 

Hence, we have added the following sentence: “resulting in the accumulation of glycosphingolipids, mainly globotriaosylceramide (Gb3) and its deacetylated form globotriaosylsphingosine (lysoGb3), the latter being commonly used as a surrogate biomarker »

We also have added the 2 references mentioned.

#2 Thank you for the proposition. We have added the 3 first references from Ortiz et al, Citro et al and Oder et al. because they illustrate well the difficulty we can meet with the management of adult Fabry patients with the problem of clinical phenotypes that we try to address in this study. We introduce them with the following sentence: “Moreover, existing scoring systems do not differentiate nonclassical from classical phenotypes of the disease whereas a growing literature suggests the need for personalized management [14–16]. »

#3 Thank you for this remark. We have added the meaning in the legend: “comparison with Mann-Whitney test p*: classical versus Nonclassical males; p**: males versus females”

---

## [Editor Report · Decision Letter 1]

6 May 2020

Cornea verticillata and acroparesthesia efficiently discriminate clusters of severity in Fabry disease

PONE-D-19-35853R1

Dear Dr. MAUHIN,

We are pleased to inform you that your manuscript has been judged scientifically suitable for publication and will be formally accepted for publication once it complies with all outstanding technical requirements.

With kind regards,

Maria Vittoria Cubellis

Academic Editor

PLOS ONE
---

## [Editor Report · Acceptance letter]

13 May 2020

PONE-D-19-35853R1 

Cornea verticillata and acroparesthesia efficiently discriminate clusters of severity in Fabry disease 

Dear Dr. MAUHIN:

I am pleased to inform you that your manuscript has been deemed suitable for publication in PLOS ONE. Congratulations! Your manuscript is now with our production department. 

With kind regards,

on behalf of

Dr. Maria Vittoria Cubellis 

Academic Editor

PLOS ONE